# FcγRs and Their Relevance for the Activity of Anti-CD40 Antibodies

**DOI:** 10.3390/ijms232112869

**Published:** 2022-10-25

**Authors:** Isabell Lang, Olena Zaitseva, Harald Wajant

**Affiliations:** Department of Internal Medicine II, Division of Molecular Internal Medicine, University Hospital Würzburg, Auvera Haus, Grombühlstrasse 12, 97080 Würzburg, Germany

**Keywords:** antibody fusion protein, CD40, CD40L, cytokine storm, FcγR receptor, immunotherapy

## Abstract

**Simple Summary:**

Targeting of CD40 with antibodies attracts significant translational interest. While inhibitory CD40 targeting appears particularly attractive in the field of organ transplantation and for the treatment of autoimmune diseases, stimulatory CD40 targeting is the aim in tumor immunotherapy and vaccination against infectious pathogens. It turned out that lack of FcγR-binding is the crucial factor for the development of safe and well-tolerated inhibitory anti-CD40 antibodies. In striking contrast, FcγR-binding is of great importance for the CD40 stimulatory capacity of the majority of anti-CD40 antibodies. Typically, anti-CD40 antibodies only robustly stimulate CD40 when presented by FcγRs. However, FcγR-binding of anti-CD40 antibodies also triggers unwanted activities such as destruction of CD40 expressing cells by ADCC or ADCP. Based on a brief discussion of the mechanisms of CD40 activation, we give an overview of the ongoing activities in the development of anti-CD40 antibodies under special consideration of attempts aimed at the development of anti-CD40 antibodies with FcγR-independent agonism or FcγR subtype selectivity.

**Abstract:**

Inhibitory targeting of the CD40L-CD40 system is a promising therapeutic option in the field of organ transplantation and is also attractive in the treatment of autoimmune diseases. After early complex results with neutralizing CD40L antibodies, it turned out that lack of Fcγ receptor (FcγR)-binding is the crucial factor for the development of safe inhibitory antibodies targeting CD40L or CD40. Indeed, in recent years, blocking CD40 antibodies not interacting with FcγRs, has proven to be well tolerated in clinical studies and has shown initial clinical efficacy. Stimulation of CD40 is also of considerable therapeutic interest, especially in cancer immunotherapy. CD40 can be robustly activated by genetically engineered variants of soluble CD40L but also by anti-CD40 antibodies. However, the development of CD40L-based agonists is biotechnologically and pharmacokinetically challenging, and anti-CD40 antibodies typically display only strong agonism in complex with FcγRs or upon secondary crosslinking. The latter, however, typically results in poorly developable mixtures of molecule species of varying stoichiometry and FcγR-binding by anti-CD40 antibodies can elicit unwanted side effects such as antibody-dependent cellular cytotoxicity (ADCC) or antibody-dependent cellular phagocytosis (ADCP) of CD40 expressing immune cells. Here, we summarize and compare strategies to overcome the unwanted target cell-destroying activity of anti-CD40-FcγR complexes, especially the use of FcγR type-specific mutants and the FcγR-independent cell surface anchoring of bispecific anti-CD40 fusion proteins. Especially, we discuss the therapeutic potential of these strategies in view of the emerging evidence for the dose-limiting activities of systemic CD40 engagement.

## 1. Introduction

### 1.1. The CD40L-CD40 System

The transmembrane receptor CD40 (Cluster of Differentiation 40) is a typical member of the tumor necrosis factor (TNF) receptor superfamily (TNFRSF). As such, its extracellular domain contains four cysteine-rich domains (CRDs), the TNFRSF defining structural element [1] (Figure 1). The receptors of the TNFRSF (TNFRs) can be categorized into three groups: TNFRs interacting with TNF receptor associated factor (TRAF) proteins, death receptors and decoy TNFRs. The latter have no own authentic signaling abilities and act as soluble or glycophosphatidylinositol (GPI)-anchored molecules to control the activity of other TNFRs by ligand competition and formation of inactive TNFR heteromers. Death receptors possess an intracellular protein–protein interaction domain, called death domain (DD), enabling the interaction with DD-containing signaling proteins and activation of cytotoxic but also proinflammatory signaling pathways [2]. CD40, however, belongs to the subgroup of TRAF interacting TNFRs which by help of short amino acid motifs recruit TRAF proteins (Figure 1), a family of signaling proteins with scaffold function and typically also E3 ligase activity [3]. CD40 directly interacts with four different members of the TRAF protein family, TRAF2, TRAF3, TRAF5 and TRAF6, and furthermore recruits TRAF1 by help of TRAF2 [4,5,6,7,8].

CD40 is primarily expressed by antigen presenting cells (APCs), such as dendritic cells (DCs), macrophages and B-cells. Furthermore, the presence of CD40 has been demonstrated on non-hematopoietic cell types such as endothelial cells, fibroblasts and smooth muscle cells. Naturally, CD40 is stimulated by CD40 ligand (CD40L, CD154, gp39), a trimeric type II transmembrane protein of the TNF superfamily (TNFSF), which is mainly expressed by activated CD4^+^ T-cells and platelets [9]. With the help of the CD40L-CD40 system, T helper cells activate APCs and thus stimulate, among other things, the formation of germinal centers in lymphoid tissues and the antibody class switch, but also the differentiation and maturation of DCs and the phagocytic activity of macrophages. Consistent with the role of the CD40L-CD40 system in antibody class switching, mutations in CD40L lead to the hyper-IgM syndrome [10].

Platelet-released CD40L leads to activation of endothelial cells and the production of chemokines, cytokines, and adhesion molecules enabling the recruitment of leukocytes [9]. The release of soluble CD40L (sCD40L) by platelets results from the proteolytic cleavage of membrane-bound CD40L (memCD40L) molecules by matrix metalloproteases. The cleavage occurs in the stalk region between the transmembrane domain and the CD40-binding TNF homology domain (THD) of the protein (Figure 1), which is also important for the trimerization of CD40L [11]. Accordingly, soluble CD40L can bind to CD40 similar to the membrane-bound form of the molecule. CD40L not only interacts with CD40, but also binds integrins, e.g., αMβ2, α5β1, α4β1, αIIbβ3 and αvβ3 [9,12]. The interaction of CD40L with integrins is non-competitive, so that ternary complexes of CD40L, CD40 and integrin can be formed. In particular, binding to α5β1 was found to be important for soluble CD40L-induced CD40-mediated stimulation of classical NFκB signaling and B-cell activation [12]. This finding implies that soluble CD40L trimers alone fail to activate CD40 comprehensively and efficiently. Indeed, it has been recognized early on that memCD40L is a more potent CD40 activator than soluble CD40L and that oligomerization of soluble CD40L trimers by genetic engineering or crosslinking antibodies strongly enhances its CD40 stimulatory activity [13,14,15,16,17,18]. In contrast, membrane-bound CD40L is an extremely potent activator of all known CD40-mediated effects. Because of the interplay between soluble CD40L and integrins, it can be very difficult to decide to what extent in vivo effects mediated by soluble CD40L are due to the activity of CD40L-CD40 or CD40L-CD40-integrin complexes.

### 1.2. Molecular Mechanisms of CD40 Activation

Irrespective of the classification into death receptors and TRAF-interacting TNFRs, the signaling competent TNFRs can also be categorized according to their response to soluble ligand molecules. One group of TNFRs, in the following category called I TNFRs, are as efficiently activated by soluble ligand trimers as by membrane-bound ligand molecules or oligomeric fusion proteins of soluble ligand molecules. An example for this type of TNFR is TNFR1. A second group of TNFRs binds soluble trimeric ligand molecules with high affinity but are not, or only to a limited extent, activated [19,20,21]. The receptors of this TNFRSF group will be referred in the following as category II TNFRs. As already mentioned above, soluble CD40L trimers have a far less CD40-stimulating effect than memCD40L or integrin-bound soluble CD40L. Thus, CD40 also belongs to the category II TNFRs.

The limited activatability of category II TNFRs seems to be due to the fact that the secondary interaction of initially formed trimeric TNFL-TNFR complexes is a crucial step in the stimulation of at least some TNFR-associated signaling pathways, especially the classical NFκB pathway (Figure 2). The latter, but also mitogen-activated protein kinase (MAPK) signaling pathways, include activation of the TRAF2-interacting E3 ligases cellular inhibitor of apoptosis 1 (cIAP1) and cIAP2 by dimerization of their really interesting new gene (RING) domain.

The aforementioned TRAF proteins assemble into homo- or heterotrimeric molecules and interact with a stoichiometry of 1:1 with liganded TNFR trimers, thus one TRAF2 trimer binds three TNFR molecules occupied by one TNFL trimer [22]. TRAF2, which has been crucially involved in the ability of CD40 to engage the classical NFκB pathway, binds a single cIAP1 or cIAP2 molecule [23,24]. A trimeric CD40L-CD40 complex thus recruits via TRAF2 only one cIAP1/2 molecule, which is insufficient for robust activation of the classical NFκB signaling pathway (Figure 2). However, if two or more trimeric CD40L-CD40 complexes are in close proximity, the cIAP1/2 molecules bound to the trimeric receptor complexes can activate each other in trans and then stimulate the classic NFκB signaling pathway (Figure 2).

TRAF6, which also interacts with CD40, acts itself as an E3 ligase catalyzing K63-ubiquitination events by virtue of its C-terminal RING domain. Again, there are structural data showing that RING domain dimerization is necessary for enzyme activation and there is evidence that intermolecular RING dimerization, also of RING domains of different TRAF protein types, is particular effective by forming large networks of RING domain dimers [25,26,27]. The latter could again explain the superior signaling activity of clustered CD40L-CD40 complexes. Many TNFRs, including CD40, autoassemble with low affinity by means of a N-terminal domain called the pre-ligand assembly domain (PLAD) [19,28,29]. In the case of CD40 and other category II TNFRs, this autoaffinity seems to be too low to promote clustering of receptor trimers liganded by soluble ligand molecules. However, the extremely high concentrations of trimeric TNFL-TNFR complexes (up to the mM range) trapped in the cell–cell contact zone between membrane TNFL-expressing and TNFR-expressing cells are sufficient to promote activating receptor clustering [19].

In accordance with the model described above, two basic principles have been identified, which enable the development of soluble CD40L variants with agonistic activity resembling memCD40L. First, the artificial enforcement of close spatial proximity of two or more trimeric CD40L-CD40 complexes. This can be achieved by physically linkage of two or more soluble CD40L trimers in similar orientation so that the CD40L-bound CD40 trimers come also in close “activating” proximity [19]. Indeed, various oligomeric fusion proteins of sCD40L have been described showing a 100 –>1000 times higher CD40 stimulatory capacity than sCD40L [13,14,16,17,18]. Second, genetic fusion of sCD40L with a protein domain which allows anchoring to a plasma membrane-resident target. If such CD40L fusion proteins are bound to their target, the high concentrations of CD40 trimers and target-bound sCD40L fusion proteins trapped in the cell–cell contact zone between CD40-expressing cells and target-expressing cells drive the “activating” clustering of these complexes. For example, we were able to show that fusion proteins of sCD40L with a scFv domain either recognizing the membrane protein “fibroblast activation protein” (FAP) or CD20, elicit strong CD40 signaling in the presence of FAP- and CD20-expressing cells [18,30].

## 2. CD40 as Therapeutic Target

Targeting of CD40 with the aim to stimulate or inhibit this receptor attracts considerable translational interest. Inhibitory CD40 targeting appears particularly attractive in the field of organ transplantation and in the treatment of autoimmune diseases [31,32,33,34]. CD40 blockade might also elicit antitumoral activity on CD40-expressing tumors. Agonistic CD40 targeting typically aims at the exploitation of the strong immunostimulatory activities of CD40 for tumor immunotherapy and vaccination against various infectious pathogens [35,36,37].

### 2.1. Inhibitory Antibody Targeting of CD40

Inhibition of CD40L-CD40 interaction can be achieved straightforwardly by conventional blocking antibodies against CD40L or CD40 (Figure 3). The important point, which has to be considered here, is to prevent binding to FcγRs and the complement activating C1q protein. The interaction with FcγRs can trigger unwanted FcγR-mediated effector functions, such as antibody-dependent cell-mediated cytotoxicity (ADCC) or antibody-dependent cellular phagocytosis (ADCP) and C1q binding can elicit complement-dependent cytotoxicity (CDC) (Figure 3).

For example, thromboembolic complications have been reported in rhesus and cynomolgus monkeys with the mouse anti-CD40L IgG2a 5C8 and the human recombinant anti-CD40L IgG1 ABI793, and early clinical trials with the anti-CD40L antibodies Ruplizumab (BG9588, humanized 5C8) and IDEC-131 (humanized IgG1, parental antibody 24–31) were terminated due to thromboembolic events observed in a phase II study with Ruplizumab [38,39,40,41,42,43], Table 1. Later studies indeed gave evidence that the thromboembolic complications are caused by immune complexes of CD40L and anti-CD40L-antibodies which activate platelets via FcγRIIA and furthermore inhibit disaggregation of platelets [44,45,46,47]. Thus, it was reported that blocking CD40L antibody variants lacking Fc effector functions did not induce thromboembolic complications [47,48]. Importantly, silencing of the Fc effector functions of anti-CD40L- and anti-CD40 antibodies leave the beneficial immunosuppressive effects of the inhibition of the CD40L-CD40 interaction intact [47,48,49,50,51]. Accordingly, the Fc-silent IgG1(N297A) anti-CD40 antibody Iscalimab was found to be well tolerated and triggered no signs of thromboembolic events in a phase I study with rheumatoid arthritis (RA) patients dosed with up to 30 mg/kg [52]; (ClinicalTrials.gov Identifier: NCT02089087, accessed on 24 October 2022). Good safety was also observed in an open-label, phase II proof-of-concept study with 15 patients treated 5 times with 10 mg/kg Iscalimab over a period of 3 months and suffering on Graves Hyperthyroidism, an autoimmune disease of the thyroid with complex symptoms [53]. Similarly, in a phase Ib study, the IgG4 anti-CD40 antibody Bleselumab was well tolerated up to a dose of 500 mg in kidney transplant recipients with standard immunosuppression [54]; (ClinicalTrials.gov: NCT01279538, accessed on 24 October 2022). In a phase II, randomized, open-label, noninferiority study, Bleselumab showed furthermore a favorable benefit-risk ratio and noninferiority in combination with immediate-release tacrolimus in kidney transplant recipients [55]; (ClinicalTrials.gov NCT01780844, accessed on 24 October 2022). In combination with mycophenolate mofetil, however, Bleselumab failed to show noninferiority [55]; (ClinicalTrials.gov NCT01780844, accessed 24 on October 2022).

### 2.2. Stimulatory Antibody Targeting of CD40

Work has been ongoing for over 20 years to develop CD40 agonists with the aim to use them as adjuvants to push vaccination against various pathogens and/or to treat tumor diseases. However, these efforts have not yet resulted in approved, clinically widely applicable CD40 agonists. This failure can mainly be attributed to three circumstances/reasons: (i) the sole use of CD40 agonists as monotherapy, (ii) the dose-limiting side effects of CD40 agonists and/or FcγR- and C1q binding (Figure 3A,B) and (iii) the insufficient activity of the CD40 agonists used.

In preclinical animal models, the sole treatment of tumors with CD40 agonists often showed very good efficacy. However, corresponding early clinical studies with CD40 agonists were not very successful and could not prove a broad therapeutic efficacy [37]. It is now assumed that in the clinic the antitumoral efficacy of CD40 agonists is dependent on a proinflammatory microenvironment, which is typically not present in advanced tumor stages. In line with this, animal studies have shown in recent years that checkpoint inhibitors or chemotherapeutic agents that promote proinflammatory processes in the tumor microenvironment have a synergistic antitumor effect with CD40 agonists [36,101]. Therefore, in the majority of the currently ongoing clinical studies with CD40 agonists, corresponding combination therapies are being investigated [36,101].

Preclinical studies in mice have identified the cytokine release syndrome (CRS) and hepatotoxicity as two major causes of anti-CD40 antibody-induced toxicity [102,103,104]. In accordance with this, the most common side effects observed in clinical trials with CD40 agonists were symptoms of the cytokine release syndrome such as fever, nausea, muscle pain and chilling, which were transient and manageable, but also release of liver enzymes and increased lipase levels [36,101]. The various clinical studies with anti-CD40 antibodies aiming at the activation of CD40 used conventional antibodies or sometimes antibodies with preference for the binding of certain FcγR types (Table 1). Importantly, clinical studies showed that anti-CD40 antibodies lacking Fc effector functions are much better tolerated than FcγR-binding competent antibody variants (Table 1). For example, the anti-CD40-IgG1 antibody Lucatumumab showed in clinical trials a maximum tolerated dose (MTD) between 3 and 4.5 mg/kg with grade 3/4 adverse effects in 32–62% of the treated patients, while its Fc-silent anti-CD40-IgG1(N297A) variant Iscalimab was well tolerated up to doses of 30 mg/kg [52,56,57,58]. The higher toxicity of FcγR-binding competent anti-CD40 antibodies clearly indicates that the FcγR-bound anti-CD40 antibodies rather than the free anti-CD40 antibody molecules are the origin of the dose-limiting activities observed in clinical trials with anti-CD40 antibodies. As discussed below in detail (see Section 2.2.1), CD40 is typically much stronger activated by FcγR-interacting anti-CD40 antibodies than by antibody molecules without FcγR binding competence. Therefore, it is difficult to attribute the dose-limiting toxicity of FcγR-/C1q-binding competent anti-CD40 antibodies to the engagement of CD40 and/or FcγR signaling and complement activation. It is thus currently unclear to which extent highly potent effector function-dead agonistic CD40 antibodies trigger the aforementioned dose-limiting effects, too. However, the preclinical studies mentioned above identified typical CD40-induced effector molecules, such as IL-12p40, TNF and IFNγ, as mediators of the toxic effects of FcγR-interacting anti-CD40 antibodies [102,104]. This suggests that CD40 signaling indeed substantially contributes to the dose-limiting toxicity of FcγR-/C1q-binding competent anti-CD40 antibodies. Therefore, it appears not unlikely that FcγR-independent CD40 agonists will also elicit dose-limiting toxicity upon systemic application thereby preventing the maximal possible CD40 activation in the tumor microenvironment. In this respect, it is worth mentioning that preclinical studies have shown that the intratumoral and systemic application of anti-CD40 antibodies is therapeutically equally effective, but that local application in the tumor is associated with fewer side effects [105,106,107,108,109]. Moreover, a recent study showed that TNF inhibition prevented the hepatotoxicity triggered by combined treatment with an anti-CD40 antibody and gemcitabine without affecting antitumor activity [110]. Thus, the maximal exploitation of potent autonomous CD40 agonists for tumor therapy may require defined treatment regimens that restrict the antibody activity to the tumor area and/or systemically inhibit CD40 effector molecules.

CD40 is strongly expressed on the surface of many hematological malignancies and CD40 expression can also be quite high on solid tumors. In line with this, early on there were also tumor therapy concepts with anti-CD40 antibodies aimed at the exploitation of Fc domain-mediated immune effector mechanisms, such as ADCC (antibody-dependent cellular cytotoxicity), CDC (complement dependent cytotoxicity) and ADCP (antibody-dependent cellular phagocytosis) to destruct CD40-expressing tumor cells (Figure 3B). In accordance with the fact, discussed below in detail, that FcγR-interacting anti-CD40 antibodies regularly acquire potent CD40-stimulatory activity (Figure 3A,B), there is furthermore evidence that such immune effector function-stimulating antibodies also trigger cell death and growth arrest by CD40 engagement. Indeed, proapoptotic CD40 effects have been described in various tumor entities [111] but antiapoptotic CD40 activities have been reported as well (e.g., [112,113]). In view of the fact that CD40 signals proliferation of non-transformed B-cells [114] and protects B-cells from cell surface immunoglobulin- and CD95-induced cell death [115,116,117], the cytotoxic activity of anti-CD40 antibodies on B-cell lymphomas is at first glance counterintuitive but could reflect that the cellular vulnerability to CD40 depends from signal strength, context and differentiation status of the cell. Indeed, CD40-induced upregulation of death ligands and the death receptor CD95 along with apoptosis induction in the absence of B-cell receptor (BCR) signaling have also been reported for non-transformed B-cells [118,119,120,121,122].

The balance between the triggering of immune effector functions (ADCC, ADCP) and CD40 signaling induced by FcγR-interacting anti-CD40 antibodies in vivo is obviously not only dependent on the availability of FcγR-expressing immune cells in the neighborhood of CD40-expressing cells but also on the FcγR type expressed by these immune cells. Dominant expression of inhibitory FcγRs could favor triggering of CD40 signaling while preferential expression of activating FcγRs could tip the balance towards cell destructive immune effector mechanisms (Figure 3A,B). Accordingly, anti-CD40 antibodies with mutations conferring preference for binding of a certain FcγR type have the potential to shape the in vivo activity of anti-CD40 antibodies towards a certain direction. For example, Horton et al. [75] introduced in a humanized IgG1 variant of the anti-CD40 antibody S2C6 two point mutations (S239D/I332E) conferring, in comparison to the non-mutated IgG1 molecule, strongly enhanced binding to all human and murine FcγRs, in particular to human FcγRIIIa and murine FcγR1 and FcγRIV. The resulting antibody XmAbCD40 and its parental IgG1 variant induced with a similar dose-response antiproliferative effects in Raji and Ramos cells but XmAbCD40 showed a significantly enhanced ability to trigger ADCC and ADCP [75]. This suggests that the particular strong increase in affinity for FcγRIIIa in XmAbCD40 has preferentially affected the ability of the antibody to trigger cell destructive immune effector functions. Vice versa, anti-CD40 antibody variants specific for murine or human CD40 harboring mutations selectively enhancing the affinity for the human inhibitory antibody FcγR2B showed strongly enhanced CD40 signaling in vitro and in FcγR2B and CD40/FcγR2B humanized mice [68,123]. Remarkably, at higher doses the FcγR2B-selective human CD40 antibody showed significant hepatotoxicity hindering tumor therapy by systemic application, but the latter could be overcome by intratumoral injection [107].

The in vivo effects of some anti-CD40 antibodies stimulating immune effector mechanisms could be further complicated by the fact that these antibodies also interfere with CD40L-CD40 interaction or modulate the activity of soluble CD40L. Therefore, at localizations where neither stimulatory nor inhibitory FcγRs are present/available they might neither destruct CD40-expressing cells nor stimulate CD40 signaling but instead block CD40 engagement by endogenous CD40L or enhance the activity of soluble CD40L (Figure 3D,E and Figure 4A). For example, the anti-CD40-IgG1 Lucatumumab (HCD122, CHIR-12.12) triggers antibody-dependent cell-mediated cytotoxicity (ADCC) but also efficiently inhibits CD40L-CD40 interaction [59,124]. While attempts to target lymphoma with this antibody aimed on the exploitation of both of these functions, a Fc-silenced form of this antibody (CFZ533) was generated to avoid ADCC and to solely block CD40-CD40L interaction for immunosuppressive treatments [50]. Indeed, as already discussed under Section 2.1., CFZ533 has been successfully used in nonhuman primates to prolong renal allograft survival without inducing B-cell depletion and Iscalimab, a fully humanized version of CFZ533, showed clinical activity in patients suffering on Graves Disease in a proof-of-concept trial and was well tolerated in a phase I study (NCT02089087) with healthy subjects and rheumatoid arthritis patients [49,52,53]. Furthermore, certain non-blocking anti-CD40 antibodies might be able to induce clustering of poorly active sCD40L-induced CD40 complexes resulting in enhanced CD40 signaling [71,78,125,126]; (Figure 3E and Figure 4A).

#### 2.2.1. Agonism of Complexes of Anti-CD40 Antibodies and FcγRs

With respect to anti-CD40 antibodies, it is extremely important to distinguish between FcγR-independent and FcγR-dependent agonistic activity, thus between the intrinsic ability of an antibody alone to trigger CD40 signaling and the ability of complexes of an antibody with FcγRs to do so. In the past 10 years, extensive in vitro and in vivo studies have given comprehensive evidence that virtually every CD40-specific antibody elicits agonistic activity when bound to Fcγ receptors [68,123,127,128]. It is worth mentioning that the agonism of FcγR-interacting anti-CD40 antibodies is independent of FcγR downstream signaling [123] and can also be realized with FcγR-transfected non-immune cells e.g., [72,128,129]. Therefore, the sheer plasma membrane-associated mode of presentation of anti-CD40 antibody molecules appears to be sufficient to constitute the agonism of anti-CD40 antibody-FcγR complexes. The nature of the FcγR type appears only to be in so far of relevance for the agonism of anti-CD40 antibodies that FcγRs differ in their affinity for the various IgG isotypes and that therefore certain anti-CD40-IgG-FcγR complexes form more efficiently than others. This issue, however, can gain overwhelming importance in vivo since the type of immune cell present in a certain tumor entity as well as the FcγR expression pattern of the various immune cell types varies considerably and the different antibody isotypes have quite different preferences for FcγRs [130,131]. Therefore, the combination of the availability of the “right” immune cell type together with the FcγR specificity of a certain anti-CD40 antibody isotype has obviously a significant impact on the achievable agonism in vivo and can explain why anti-CD40 antibodies show quite different in vivo performance ranging from antagonism over model-dependent quantitatively widely differing agonism despite having a comparable FcγR-dependent agonistic activity in vitro.

Several groups have shown that bispecific anti-CD40 antibody variants that recognize plasma membrane-associated targets distinct from CD40 elicit up to a 1000-fold increased CD40-stimulating activity after binding to this second antigen [129,132,133,134,135]. These studies not only demonstrate that the agonism manifesting anti-CD40 antibody-FcγR interaction can be replaced by molecularly different interactions emphasizing the relevance of the plasma membrane-associated presentation mode for agonism, but also offers the opportunity to prevent systemic CD40 activation by addressing a selectively expressed target, e.g., a tumor antigen.

It appears quite plausible that the agonism of FcγR binding-competent anti-CD40 antibodies is due to the same molecular mode of action that also applies to the much stronger CD40-stimulating activity of membrane CD40L compared to soluble CD40L trimers. Namely, the presence of high local concentrations of plasma membrane agonist-bound CD40 molecules (FcγR-anti-CD40-antibody-CD40 dimers) in the cell–cell contact zone between CD40^+^ cells and FcγR^+^ cells favoring secondary clustering to fully active oligomeric agonist-CD40 complexes (Figure 2).

In view of the fact that anti-CD40 antibodies bound to FcγRs regularly display strong agonism, it is evident that the clinical development of in vivo antagonistically acting anti-CD40 antibodies is de facto only possible with antibody isotypes that do not or only very slightly interact with FcγRs, or with immunoglobulin mutants with defective FcγR binding (e.g., IgG1-N297A or IgG1-LALA).

#### 2.2.2. Problems and Limitations of CD40 Engagement by FcγR-Interacting Anti-CD40 Antibodies

In general, it must be considered that it is typically not possible to achieve activation of all CD40 molecules with anti-CD40 antibodies in vivo, due to the limited availability of FcγR molecules. For example, anti-CD40-mIgG1 antibodies stimulate significant proliferation of B-cells from wild-type mice but not of B-cells from FcγRIIB-deficient mice [128]. However, this FcγRIIB-dependent CD40 agonism can be further increased by one to two orders of magnitude in the wild-type and FcγRIIB-deficient B-cells if FcγR-expressing transfectants are added [128]. Apparently, the physiological FcγR expression levels of B-cells are not sufficient to allow occupancy of all CD40 molecules of the B-cells with FcγR binding-competent anti-CD40 antibody molecules. Furthermore, the binding of anti-CD40 antibodies to activating FcγRs not only empower these antibodies to efficiently activate CD40 but, as discussed already before, can also result in the destruction of the CD40-expressing target cells by effector functions of the FcγR-expressing cells. Finally, conventional anti-CD40 antibodies have to compete with endogenous IgG molecules for FcγR binding, resulting in the need to apply high anti-CD40 antibody doses to reach relevant FcγR occupation.

#### 2.2.3. Anti-CD40 Antibodies with Intrinsic Thus FcγR-Independent Agonism

The majority of reports on agonistic anti-CD40 antibodies investigated CD40 agonism in FcγR-expressing cell types (DCs, B-cells) or observed enhanced anti-CD40 agonism upon crosslinking with secondary antibodies but nevertheless imprecisely attributed the agonism solely to the anti-CD40 antibody and not to the FcγR-bound antibodies or the anti-CD40-anti-IgG complexes. Thus, many published “agonistic” anti-CD40 antibodies have no or only extremely moderate intrinsic agonistic activity.

However, some studies have explicitly demonstrated robust intrinsic autonomous agonism of anti-CD40 antibodies, especially for anti-CD40 antibodies of the human IgG2 isotype (hIgG2) [136]. Interestingly, the agonistic activity of anti-CD40-hIgG2 antibodies has been assigned to isoform B of the hIgG2 isotype, which differs from the A isoform of the hIgG2 molecule in the formation of disulfide bridges between the CH1 and CL domains, and has a less flexible arrangement of the two Fab domains of the antibody [137,138,139,140]. Anti-CD40-hIgG2 antibodies that have mutations that produce either only isoform A (e.g., HC-C233S) or only isoform B (e.g., HC-C127S or LC-C214S/HC-C233S) therefore elicit no agonistic activity or even show an increased FcγR-independent agonism compared to the parental hIgG2 molecule [136]. In line with the two-step model of CD40 activation described in Figure 2, it has been found that most anti-CD40-IgG2B antibodies, in contrast to their IgG1 counterparts, indeed autonomously trigger strong CD40 clustering [141]. However, it is unclear, whether the secondary clustering of initially formed CD40-IgG2B complexes is powered by CD40-CD40 or IgG2B-IgG2B interactions (Figure 4C). It is also worth mentioning that the FcγR-independent agonism of anti-CD40-hIgG2 or anti-CD40-hIgG2B antibodies seems to still be significantly lower than that of FcγR binding-competent anti-CD40 antibodies in the presence of FcγRs or have been challenged for its relevance in the human system [68,128].

## 3. Conclusions and Perspective

There is now comprehensive and compelling evidence that the effects of anti-CD40 antibodies on CD40 expressing cells are in vivo crucially dependent on their ability to interact with FcγRs and C1q or not. Thus, to exploit anti-CD40 antibodies blocking CD40L binding as specific CD40 antagonists, undesired FcγR/C1q-mediated activities have to be prevented by the use of antibody isoforms or antibody mutants devoid of FcγR/C1q binding. Engagement of CD40 signaling by anti-CD40 antibodies, on the other hand, typically requires plasma membrane-associated presentation by Fcγ receptors (FcγRs). To avoid the destruction of the targeted CD40-expressing cells in this case, preferential antibody interaction with the inhibitory FcγR2B is aspired. Indeed, the insufficient consideration of the relevance of FcγR/C1q-binding for the effects of anti-CD40 antibodies on CD40 activities can explain why early clinical studies aimed at inhibition or activation of CD40 were less successful. The ongoing clinical activities with optimized anti-CD40 antibodies devoid of FcγR/C1q binding and FcγR2B-preference now have proof of the therapeutic promise of antagonistic and agonistic CD40 targeting. There is initial preclinical evidence that off-tumor activities arising from systemic CD40 activation can limit the success and potential of agonistic CD40 targeting. Limitations arising from off-tumor CD40 engagement, however, might become manageable by local application or use of bispecific CD40 antibodies with conditional agonism.

The relevance of FcγR-binding for agonistic in vivo activity is certainly best studied for anti-CD40 antibodies but there is also a considerable number of publications reporting FcγR-dependent agonism for antibodies targeting other category II TNFRs, including 4-1BB, CD27, CD95, DR5, Fn14, OX40 and TNFR2 [19,142]. Therefore, the experiences made in the clinical development of anti-CD40 antibodies with respect to the relevance of FcγR/C1q binding have the potential to guide the development of other category II TNFR antibodies for clinical use.

## Figures and Tables

**Figure 1 ijms-23-12869-f001:**
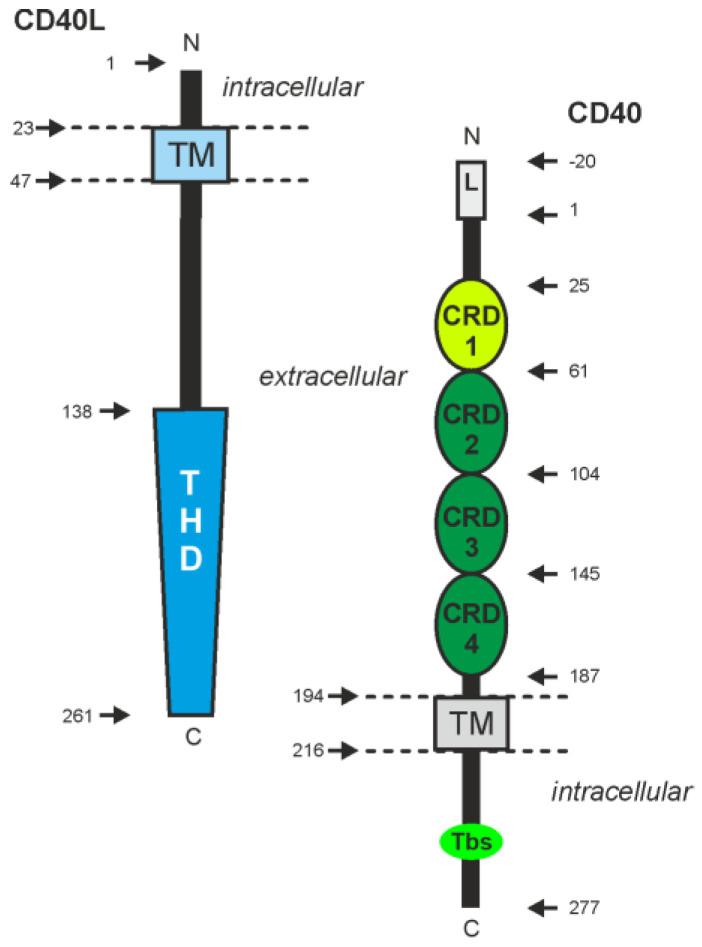
Domain architecture of CD40 and its ligand CD40L/CD154. CRD1 to CRD4 define CD40 as a TNFR. CRD1 is also functionally defined as pre-ligand binding assembly domain (PLAD), which mediates low-affinity CD40 self-assembly in the absence of CD40L. The TRAF binding site (Tbs) consisting of the amino acid motif PVQET is shown in overproportional size. The THD (TNF homology domain) defines CD40L as a member of the TNFSF. Arrows indicate amino acid positions according to the mature full-length proteins.

**Figure 2 ijms-23-12869-f002:**
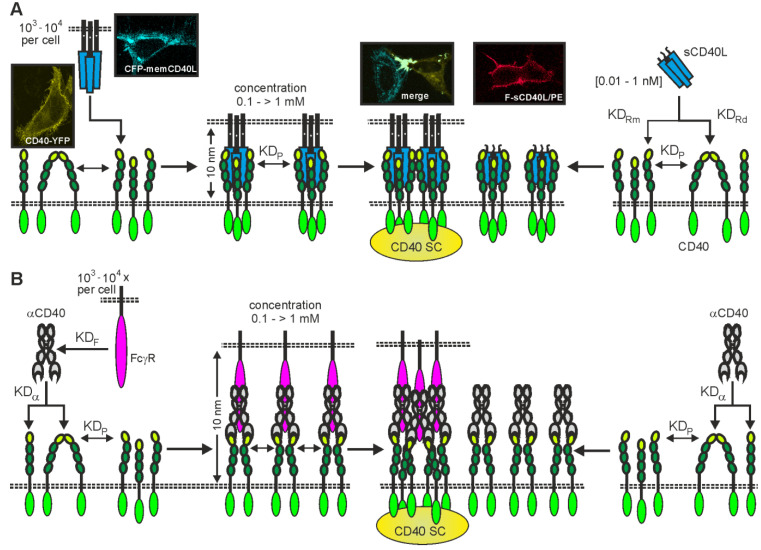
Two-step model of CD40 activation by CD40L (**A**) and anti-CD40 antibodies (**B**). In a first step, CD40L and anti-CD40 antibodies (αCD40) bind to monomeric (KD_Rm_ or KD_α_) or dimeric CD40 molecules (KD_Rd_ or KD_α_) with high affinity. The latter are formed to a small extent due to the weak autoaffinity of the CD40 pre-ligand binding assembly domain (PLAD; KD_P_). However, the trimeric (**A**) and dimeric CD40 complexes (**B**) resulting from CD40L and αCD40 binding are not sufficient, for example, to activate the classic NFκB signaling pathway. The latter requires the interaction of two (or more) trimeric TRAF adapter proteins and thus CD40 complexes with six or more receptor molecules, which can form in a second step by autoaggregation of CD40L- or αCD40-bound CD40 complexes. Due to the several powers of 10 increased local concentrations of trimeric and dimeric CD40 complexes that form in the cell–cell contact zone between CD40^+^ cells and membrane CD40L^+^ (**A**) or FcγR^+^ cells (**B**), respectively, membrane CD40L and FcγR-αCD40 complexes display a far stronger agonism than soluble CD40L and free αCD40 molecules. The micrographs shown in A illustrate the aggregation behavior of CD40, membrane CD40L, and CD40 in complex with memCD40L or soluble CD40L by help of fluorescent fusion proteins (CD40-YFP; CFP-memCD40L) or staining of Flag-tagged soluble CD40L with PE-conjugated anti-Flag antibody M2 (F-sCD40L). CD40 SC, CD40 signaling complex. Dotted lines indicate plasma membranes.

**Figure 3 ijms-23-12869-f003:**
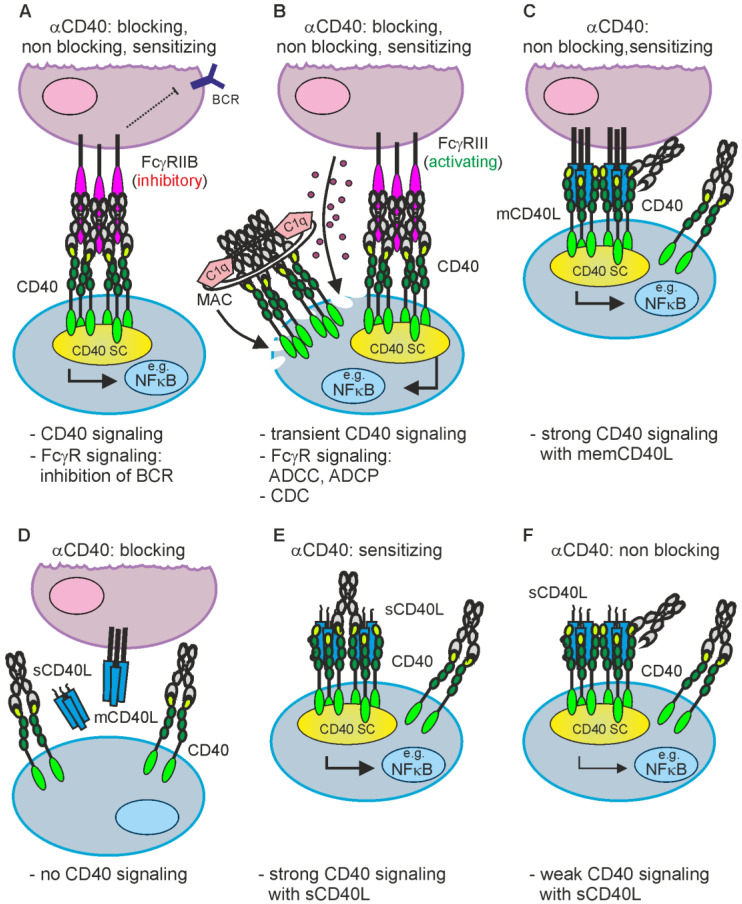
Isotype and the effect on CD40L-CD40 interaction determine the possible mode of actions of anti-CD40 antibodies. (**A**,**B**) Irrespective of the effect on CD40L-CD40 interaction, anti-CD40 antibodies of the appropriate isotype can stimulate inhibitory (**A**) or activating FcγRs (**B**) but also complement-mediated cell lysis via C1q binding and formation of the membrane attack complex (MAC). (**B**) FcγR binding by anti-CD40 antibodies further results typically in strong CD40 engagement. (**C**,**D**) Non-blocking and sensitizing anti-CD40 antibodies have no modulating effect on memCD40L-induced CD40 activation (**C**) while blocking antibodies completely prevent CD40 engagement by CD40L (**D**). (**E**,**F**) Sensitizing, non-blocking anti-CD40 antibodies enhance the activity of soluble CD40L, while non-blocking antibodies leave the weak CD40 signaling triggered by sCD40L intact. CD40 SC, CD40 signaling complex.

**Figure 4 ijms-23-12869-f004:**
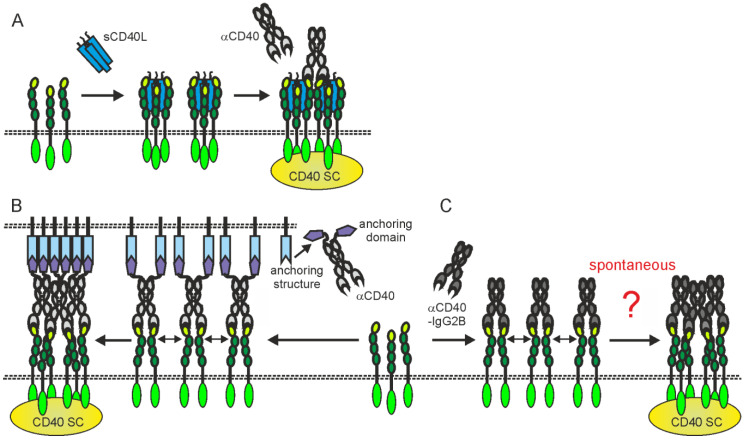
Activating CD40 clustering by (**A**) mixtures of sCD40L and non-blocking anti-CD40 antibodies, (**B**) anti-CD40 antibody fusion proteins with an anchoring domain enabling binding to a plasma membrane-localized anchoring structure or (**C**) anti-CD40-hIgG2 antibodies. Dotted lines indicate plasma membranes. For details, please see text.

**Table 1 ijms-23-12869-t001:** Characteristics of prominent preclinical and clinical anti-CD40 antibodies. MTD, maximum tolerable dose; DLT, dose-limiting toxicity; GD, Graves disease; MM, multiple myeloma; CLL, chronic lymphocytic leukemia; NHL, non Hodgkin lymphoma; RA, rheumatoid arthritis.

Antibody	Isotype	Effect on CD40L Binding	Intended Applications	Activity on/with FcγR^+^ Cells	FcγR-Independent Agonism	Remarks	Ref.
Lucatumumab HCD122 CHIR-12.12	Fully human IgG1	Blocking	Tumor therapy	ADCC	No	B-cell depletion Improved renalallograft survival (app. 50 days) MTD: 4.5 mg/kg (MM) MTD: 3 mg/kg (CLL) MTD: 4 mg/kg (NHL + HL)	[49,56,57,58,59]
Iscalimab CFZ533 HCD122(N297A)	IgG1 with N297A mutation	Blocking	RA, GD, Transplantation	No	No	Improved renalallograft survival (>100 days) Suppression of GC development 3–30 mg/kg, all doses safe and well tolerated (healty, RA, GD)	[49,52,53,60]
2C10R1 and 2C10R4	IgG1 (2C10R1) IgG4 (2C10R4)	Blocking		No agonism	No	Improved islet graft survival Prolonged cardiac xenoplant survival	[61,62]
KPL-404 (humanized 2C10R4)	IgG4 with S228P mutation	Blocking		No agonism	No	10 mg/kg i.v. no obvious safty findings in cynomolgus but inhibition of CD40 signaling	[63,64]
Selicrelumab CP-870,893	IgG2	Non blocking	Tumor therapy	Agonistic on DCs	Yes	MTD: 0.2 mg/kg (solid cancer)	[65,66,67]
2141-V11 (FcγR2B-enhanced CP-870,893)	IgG1 with G237D-P238D-H268D-P271G-A330R mutations		Tumor tharapy	Agonistic	Not tested	Better antitumor activity than CP-870-893 in hCD40/hFcγR mice Intratumoral injection, systemic caused liver damage	[68,69]
Ravagalimab ABBV-323 (enhanced FcRn binding)	IgG1 with L234A-L235A-T250Q-M428L mutations	Blocking	Ulcerative colitis Sjogren’s syndrome	No	No	IgG2 variant of Abbv-323 transform it to an agonist	[70,71]
Sotigalimab APX005M (preferential FcγR2B binding)	IgG1 with S267E mutation	Blocking	Tumor therapy	High agonism but no ADCC	No	Recommended dose 0.3 mg/kg	[72,73]
CDX-1140	IgG2	Non blocking	Tumor therapy	Agonistic on DCs and B-cells	Yes but synergisitic with soluble CD40L		[74]
XmAbCD40 (humanized S2C6 with enhanced FcγR binding)	IgG1 with S239D-I332E mutations		Tumor therapy	ADCC, ADCP, CDC		Antitumoral in xenogeneic tumor models	[75]
Mitazalima ADC-1013 JNJ-64457107 (phage display improved B44)	IgG1		Tumor therapy	Agonistic on DCs, ADCC	No	Antitumoral in xenogeneic tumor models 0.4 mg/kg intratumoral 0.075 mg/kg iv	[76,77]
Bleselumab ASKP1240 341 4D11	Fully human IgG4	Blocking	Transplantation	No		Kidney transplant recipients up to 500 mg	[54]
341-IgG2	IgG2 variant of Bleselumab	Blocking	Tumor therapy		Yes	IgG2 variant of Bleselumab act as an agonist	[71]
BI 655064		Blocking				120 mg well tolerated	[78]
ChiLob7/4	IgG1			ADCC, CDC, agonistic activity	Cross-linked than agonistic	MTD: 2–3 mg/kg	[79,80]
SGN-14	mIgG1	Non blocking	Tumor therapy		Synergisitic with soluble CD40L		[81]
Dacetuzumab, SGN-40 (humanized SGN-14)	IgG1	Non blocking	Tumor therapy	ADCC, ADCP	Partial agonisitic	8 mg/kg modest activity and acceptable toxicity in DLBCL patients	[82,83,84]
ch5D12, mu5D12	mIgG2b	Blocking ^a^		(with human FcγRII cells)			[85]
PG102	Less immunogeneic form of ch5D12	Blocking		Poor	No	Antagonizes CD40L-induced CD40 signaling but degrades TRAF proteins	[86]
3A8	mIgG2b	Non Blocking but inhibitory ^a^		(with human FcγRII cells) no/partial agonism with B-cells	No	Prolongs islet allograft survival in rhesus macaques	[85,87]
5C11	mIgG1		Tumor therapy	DC maturation			[88]
B44	IgG1	Non blocking		Agonisitic on B-cells			[89]
3G3				Inhibit MLR			[90]
G28.5	mIgG1	artly Blocking		Agonistic on B-cells (with human FcγRII cells)			[91,92,93,94]
Chi220 BMS-224819	IgG1	Blocking	Transplantation	Agonisitic on B-cells		B-cell depletion in vivo	[95,96]
626.1	mIgG1			Agonistic on B-cells	Fab2 agonistic on B-cells		[97,98]
MAB89	mIgG1	Blocking		Agonistic on B-cells			[91,99]
17.40	mIgM	Blocking		Agonistic on B-cells		Synergize with MAB89 and S2C6 but not sCD40L	[91]
3C6	mIgG2b	Blocking ^a^		(with human FcγRII cells)	No		[85]
S2C6	mIgG1	Partly blocking		Agonistic on B-cells		Enhanced by anti-mIgG	[91,100]

^a^ Inhibition of B-cell proliferation by memCD40L-expressing EL45B cells.

## Data Availability

Not applicable.

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
