# Peer review of "FcγRs and Their Relevance for the Activity of Anti-CD40 Antibodies"

_ijms, 2022, doi:10.3390/ijms232112869_

Round 1
Reviewer 1 Report
Congratulations on the very professionally written review.
Author Response
Comments and Suggestions for Authors
Congratulations on the very professionally written review.
We are very pleased about the positive response to our manuscript and thank for taking time for reviewing.
Reviewer 2 Report
In this review Lang et al review the importance of FcgRs in the activity of anti-CD40 antibodies. In general it is comprehensive, providing a good level of background regarding CD40 structure and biology. However some parts are either incorrect or require clarification as detailed below.
l Line 177-180 and also the abstract: interaction with FcgRs is not known to trigger CDC. CDC is mediated by interaction of the antibody Fc (of the appropriate isotype) with C1q. This and other references relating to CDC need correcting.
l Direct evidence for the ADCC and ADCP mediating properties of anti-CD40 mAb should be provided. i.e. in vivo has this been shown to occur as opposed to in vitro in ADCC/ADCP assays. If primary research demonstrating this is not provided then the statements should be refined to indicate this is a possibility indicated from in vitro assays.
l For Figure 1 legend, “The TRAF binding site (Tbs) is shown as a green oval in overproportional size.” The description of Tbs is unclear because all ovals are green and it is clearly not overproportional in size. Also, after “THD” should there be a “C” to indicate C-terminus?
l Line 93-102: the authors class TNFRs into group I and II based on the ability of soluble ligand trimers to potently activate TNFRs. However, the authors only provided one reference (i.e. ref 13) which is also a review article. When different research groups study soluble ligands, trivalent and higher order valency, the methods for producing recombinant ligands differ widely, such as through non-covalent trimerization motif and covalent multimeric fusion constructs. Therefore it is sometimes difficult to draw firm conclusions from these studies when comparing soluble ligand vs membrane-bound ligand activity. The authors should expand on the literature evidence to provide firm support to the concept of group I and II TNFRs.
l Line 246-248: authors use the phrasing “FcgR-bound anti-CD40 antibodies”, there is a lack of direct experimental data showing that anti-CD40 antibodies actually bind FcgRs in vivo, it is perhaps more accurate to say “FcgR-competent anti-CD40 antibodies”. Same as in line 244 and elsewhere where this phrase is used.
l Line 254-256: the meaning of this sentence is unclear.
l Line 380-381: “maximum 380 possible level of CD40 activation”, what is considered maximum?
l The article seems to lack a final overall summary/conclusion and ends abruptly
Minor: in places the English could be improved. E.g.
Line 93 “Irrespective of the classification in death receptors and TRAF-interacting TNFRs” – presumably the authors mean “into”
Line 99: “or only very limited activated this way”
Line 167 “Stimulatory and inhibitory targeting of CD40 attract huge translational interest
Several typos or sentences/statement that arte unclear in Table 1 “Cross-linked than agonistic;” ; “IgG1(S267E) FcR2B prefrence”; “Humanized IgG1 of phage display improved B44 “; “partial agonisitic”; “agonisitic”
Line 347 “FcRspecificity”
Line 353 Some sentences are unclear and should be clarified “that the sole presentation in plasma membrane-associated form is the cue that converts an inactive or poorly active “CD40 binder” into a strong agonist "
Author Response
First of all, we want to thank for taking time for reviewing and the helpful comments.
Line 177-180 and also the abstract: interaction with FcgRs is not known to trigger CDC. CDC is mediated by interaction of the antibody Fc (of the appropriate isotype) with C1q. This and other references relating to CDC need correcting.
We apologize for this mistake and corrected the corresponding incorrect phrases throughout the manuscript (e.g. lines 178, 181, 220, 252, 254, 259-261).
Direct evidence for the ADCC and ADCP mediating properties of anti-CD40 mAb should be provided. i.e. in vivo has this been shown to occur as opposed to in vitro in ADCC/ADCP assays. If primary research demonstrating this is not provided then the statements should be refined to indicate this is a possibility indicated from in vitro assays.
The evidence for the relevance of ADCC/ADCP for anti-CD40 effects in vivo is indirect and is mainly based on the comparison of ADCC/ADCP competent and incompetent antibodies with the same idiotype. We clarified/specified this in the revised manuscript (e.g. lines 195, 245, 318) .
For Figure 1 legend, “The TRAF binding site (Tbs) is shown as a green oval in overproportional size.” The description of Tbs is unclear because all ovals are green and it is clearly not overproportional in size. Also, after “THD” should there be a “C” to indicate C-terminus?
“Overproportional” refers to the fact that the Tbs comprises only 5 amino acids. Thus, the “Tbs” oval is oversized in comparison to the other parts of the receptor. Since the “Tbs” oval is already labeled with “Tbs”, we deleted the unnecessary color information. The corresponding phrase in the figure legend is now: “The TRAF binding site (Tbs) consisting of the amino acid motif PVQET is shown in overproportional size.“ We also included the missing “C” to indicate the C-terminus of CD40L as suggested.
Line 93-102: the authors class TNFRs into group I and II based on the ability of soluble ligand trimers to potently activate TNFRs. However, the authors only provided one reference (i.e. ref 13) which is also a review article. When different research groups study soluble ligands, trivalent and higher order valency, the methods for producing recombinant ligands differ widely, such as through non-covalent trimerization motif and covalent multimeric fusion constructs. Therefore it is sometimes difficult to draw firm conclusions from these studies when comparing soluble ligand vs membrane-bound ligand activity. The authors should expand on the literature evidence to provide firm support to the concept of group I and II TNFRs.
In the mentioned review (ref.19), we comprehensively cite the original literature describing the superior receptor-stimulating activity of oligomeric and cell surface-anchored soluble ligand trimers. We also explicitly cite in our manuscript the key references showing this specifically for CD40L (refs 13-18). Although not in all these original publications the activity of the oligomerized/anchored trimers have been benchmarked with the activity of membrane-bound ligand, it is obvious of the literature as a whole that the oligomerized/anchored trimers elicit largely similar effects as the biological active memTNFL molecules. We add two additional reviews from other groups (refs. 20,21) covering this issue. In view of the bulk of original literature addressing the different responsiveness of certain TNFR family members to soluble and membrane-bound TNFLs activity, we feel it is reasonable to cite reviews for further reading.
l Line 246-248: authors use the phrasing “FcgR-bound anti-CD40 antibodies”, there is a lack of direct experimental data showing that anti-CD40 antibodies actually bind FcgRs in vivo, it is perhaps more accurate to say “FcgR-competent anti-CD40 antibodies”. Same as in line
Where appropriate, we replaced “FcgR-bound” by phrases such as “FcgR-interacting” and “FcgR binding-competent”. In line 244 the phrase was used in combination with “…indicates”, thus as a conclusions, and appears therefore correct to us.
l Line 254-256: the meaning of this sentence is unclear.
Rephrased for better understanding.
l Line 380-381: “maximum 380 possible level of CD40 activation”, what is considered maximum?
Rephrased to “…to achieve activation of all CD40 molecules with anti-CD40 antibodies in vivo, due to the limited availability of…” for better understanding.
l The article seems to lack a final overall summary/conclusion and ends abruptly
We included a final Conclusion and Perspective paragraph (lines 426-451).
Minor: in places the English could be improved. E.g.
Line 93 “Irrespective of the classification in death receptors and TRAF-interacting TNFRs” – presumably the authors mean “into”
Corrected
Line 99: “or only very limited activated this way”
Changed to “…are not, or only to a limited extent, activated.”
Line 167 “Stimulatory and inhibitory targeting of CD40 attract huge translational interest
Rephrased to: “Targeting of CD40 with the aim to stimulate or inhibit its activity attracts considerable interest”
Several typos or sentences/statement that arte unclear in Table 1 “Cross-linked than agonistic;” ; “IgG1(S267E) FcgR2B prefrence”; “Humanized IgG1 of phage display improved B44 “; “partial agonisitic”; “agonisitic”
Statements/remarks in Table 1 were checked and improved.
Line 347 “FcgRspecificity”
Corrected
Line 353 Some sentences are unclear and should be clarified “that the sole presentation in plasma membrane-associated form is the cue that converts an inactive or poorly active “CD40 binder” into a strong agonist "
This sentence/conclusion has been deleted here because the point is better explained a few lines below. Some other sentences were rephrased for better readability.
Reviewer 3 Report
The manuscript described the CD40/CD40L system and the use and limitations of anti CD40 antibodies. CD40 and CD40L are important for the fine-tuning of immune cell activity. Under physiologic conditions, D40 is important for T-cell mediated activation of B cells and the production of class-switched antibodies.
Under pathologic conditions where two scenarios are possible. Patient with an autoimmune disease need a down-regulation of immune cell activation, while in inflammatory conditions or after transplantation less activity is needed. Therefore, is should be possible to use CD40 antibodies context-dependently if the mechanisms of their molecular regulation are understood.
The manuscript is well written and clearly structured. The immunologic background is thoroughly explained and the figures are helpful to understand the complex layers of molecular regulations. Therefore I only have 2 small comments:
chapter 2.1: The interaction of the described blocking antibodies with Fcgamma receptors is stated to result in ADCC, CDC and ADCP, all of which are not wanted under therapeutic conditions. However, this is would not restricted to anti-CD40 antibodies but a general point. Is this problem more frequently or more strongly observed for CD40 antibodies?
Line 323: define graves disease so that the link to autoimmunity gets clear
Could you finish with a summary or perspective that states where we are in terms of current clinical use and further improvements?
Author Response
The manuscript is well written and clearly structured. The immunologic background is thoroughly explained and the figures are helpful to understand the complex layers of molecular regulations. Therefore I only have 2 small comments:
We are very pleased about the positive response to our manuscript and thank for taking time for reviewing.
chapter 2.1: The interaction of the described blocking antibodies with Fcgamma receptors is stated to result in ADCC, CDC and ADCP, all of which are not wanted under therapeutic conditions. However, this is would not restricted to anti-CD40 antibodies but a general point. Is this problem more frequently or more strongly observed for CD40 antibodies?
This issue is certainly best studied for anti-CD40 antibodies but FcgR-dependent agonism have also been reported for several other antibodies targeting category II TNFRs, such as 4-1BB, CD27 etc. We addressed this in the new Conclusion and Perspective paragraph (lines 445-451).
Line 323: define graves disease so that the link to autoimmunity gets clear
Many thanks for this suggestion. We adressed this point in lines 200-201 of the revised manuscript.
Could you finish with a summary or perspective that states where we are in terms of current clinical use and further improvements?
We covered this issue now in a brief and final Conclusion and Perspective paragraph (lines 426-451).